# Management and Utilization of Plant Genetic Resources for a Sustainable Agriculture

**DOI:** 10.3390/plants11152038

**Published:** 2022-08-04

**Authors:** Ranjith Pathirana, Francesco Carimi

**Affiliations:** 1Plant & Food Research Australia Pty Ltd., Waite Campus Research Precinct—Plant Breeding WT46, University of Adelaide, Waite Rd, Urrbrae, SA 5064, Australia; 2School of Agriculture, Food and Wine, Waite Campus Research Precinct—Plant Breeding WT46, University of Adelaide, Waite Rd, Urrbrae, SA 5064, Australia; 3Istituto di Bioscienze e BioRisorse (IBBR), C.N.R., Corso Calatafimi 414, 90129 Palermo, Italy

**Keywords:** centres of origin, crop wild relatives, crop domestication, cryopreservation, genebank, conservation, in vitro storage, germplasm, ecosystem restoration, plant breeding, climate change

## Abstract

Despite the dramatic increase in food production thanks to the Green Revolution, hunger is increasing among human populations around the world, affecting one in nine people. The negative environmental and social consequences of industrial monocrop agriculture is becoming evident, particularly in the contexts of greenhouse gas emissions and the increased frequency and impact of zoonotic disease emergence, including the ongoing COVID-19 pandemic. Human activity has altered 70–75% of the ice-free Earth’s surface, squeezing nature and wildlife into a corner. To prevent, halt, and reverse the degradation of ecosystems worldwide, the UN has launched a Decade of Ecosystem Restoration. In this context, this review describes the origin and diversity of cultivated species, the impact of modern agriculture and other human activities on plant genetic resources, and approaches to conserve and use them to increase food diversity and production with specific examples of the use of crop wild relatives for breeding climate-resilient cultivars that require less chemical and mechanical input. The need to better coordinate in situ conservation efforts with increased funding has been highlighted. We emphasise the need to strengthen the genebank infrastructure, enabling the use of modern biotechnological tools to help in genotyping and characterising accessions plus advanced ex situ conservation methods, identifying gaps in collections, developing core collections, and linking data with international databases. Crop and variety diversification and minimising tillage and other field practices through the development and introduction of herbaceous perennial crops is proposed as an alternative regenerative food system for higher carbon sequestration, sustaining economic benefits for growers, whilst also providing social and environmental benefits.

## 1. Introduction

With the global population expected to reach 9 billion by the middle of this century and the land area available for food production stagnating, or even reducing, the challenge of global food security is ever increasing. Almost one out of every nine people in the world suffers from hunger, and the number of hungry people is growing, albeit slowly [1]. The 2030 Agenda for Sustainable Development puts forward a transformational vision recognizing that our world is changing, bringing with it new challenges that must be overcome if we are to live in a world without hunger, food insecurity, and malnutrition in any of its forms. More than 820 million people in the world go hungry today, up from 784 million in 2015, emphasising the immense challenge of achieving the United Nations Zero Hunger target by 2030 [2]. Hunger is rising in almost all subregions of Africa and, to a lesser extent, in Latin America and Western Asia. It is heartening to see progress in Southern Asia in the last 5 years, but the prevalence of undernourishment in this subregion is still the highest in Asia. Another disturbing scenario is the fact that about 2 billion people in the world experience moderate to severe food insecurity, thus experiencing greater risk of malnutrition and poor health. With the drop in economic growth, food access disruptions, increasing unemployment, rising food costs, and exacerbated poverty, food insecurity will affect a further 83–132 million people [3].

Concurrently with some regions of the global population increasingly experiencing hunger, around one million animal and plant species are threatened with extinction purely because of human activity—many within decades [4]. Even though natural ecosystem services are critical for our survival—providing oxygen, regulating weather patterns, pollinating our crops, and providing us with food, fibre, and feed for livestock—human activity has altered 70–75% of the global ice-free Earth’s surface [5], squeezing natural ecosystems into an ever-decreasing corner of the planet. The health of ecosystems on which we and all other species depend is deteriorating more rapidly than ever, affecting the very foundations of our economies, livelihoods, food security, health, and quality of life worldwide.

Using a data harmonisation procedure to reduce uncertainties in satellite-based land cover maps, Song [6] estimated the annual value of the world’s total terrestrial ecosystem services at USD 49.4 trillion. Land and its biodiversity also represent essential, intangible benefits to humans, such as cognitive and spiritual enrichment, a sense of belonging, and aesthetic and recreational values. Valuing ecosystem services with economic methods often overlooks these intangible services that shape societies, cultures, and quality of life, as well as the intrinsic value of biodiversity. The Earth’s land area is finite. Using land resources sustainably is fundamental to human well-being. Despite commitments made in 2010, biodiversity has further declined in the past decade [7]. To prevent, halt, and reverse the degradation of ecosystems worldwide, the UN has launched a Decade of Ecosystem Restoration (2021–2030). This action plan is globally coordinated through the governing body of the Convention on Biological Diversity and is in response to a call from scientists, as articulated in the Special Report on Climate Change and Land of the Intergovernmental Panel on Climate Change, the Rio Conventions on Climate Change and Biodiversity, and the UN Convention to Combat Desertification. This new draft for the post-2020 Global Biodiversity Framework comprises 21 targets and 10 milestones [8].

The Green Revolution aimed to resolve the global food crisis through breeding cultivars for mechanized monocultures with high inputs of pesticides and fertilizers. Although the increased yield helped to save the cultivation of an estimated 17.9–26.7 million hectares of new land under crops [9], the resulting increase in food production during the past five decades was accompanied by environmental degradation and deficiency in micronutrients in populations [10]. Furthermore, there is strong evidence that modern farming practices and intensified systems may be linked to disease emergence and amplification. There are many examples of zoonotic disease emerging at the wildlife–livestock–human interface which are associated with agricultural intensification and environmental change, such as habitat fragmentation and ecotones, reduced biodiversity, agricultural changes, and increasing human density in ecosystems [11,12], including Ebola virus [13,14] and recently, COVID-19 [15]. It is now clear that, as predicted by Borlaug in his acceptance speech for the Nobel Peace Prize (1970), further actions are needed to resolve the worsening environmental and social crises limiting food production and planetary health.

We have seen unprecedented levels of global warming since the beginning of the industrial era. Modelling has shown that each degree in temperature rise would cause a drop in crop production of 7.4% for corn, 6% for wheat, 3.2% for rice, and 3.1% for soybean [16]. This is significant because these four crops provide two-thirds of the human caloric intake. It is not only production, but also the quality of food that is at stake. Increased temperature and, to a lesser extent, increased atmospheric CO_2_, will affect soil biogeochemical processes by altering microbial community dynamics and activity, and geochemical reactions. This will result in alterations of ionic composition of the rhizosphere, hence affecting their uptake by crop plants. For example, it has been shown that future conditions will likely lead to a greater proportion of the more toxic form of arsenic, pore-water arsenite, in the rhizosphere of Asian and Californian rice soils. Simulating these conditions under controlled environmental conditions, Muehe et al. [17] showed that elevated temperatures not only reduced rice yield by 39%, but also increased the amount of inorganic arsenic twofold in rice grain. Arsenic accumulation in rice is already becoming a problem in many South and Southeast Asian countries [18,19]. Therefore, agriculture requires more resilient varieties that can address likely problems induced by climate change; germplasm collections hold the key to developing such cultivars.

In broad terms, germplasm is the diversity of a cultivated species and its wild relatives that can hybridise and produce fertile progeny. Germplasm is of great importance for plant breeding because it carries genes that have potential value for improving crop yield, quality, and adaptation to the environment, including biotic and abiotic stresses. Thus, recognising the increasing frequency of natural disasters affecting the agricultural sector and their impact on food security, the introduction of longer-term measures to increase farmers’ and households’ resilience to natural disasters and climate change, such as the promotion of drought-tolerant crops and varieties, and livelihood diversification, is important. It is in this context that this paper on genetic diversity of cultivated species is presented, discussing the current situation and possible response options to contribute positively to sustainable development.

## 2. Origin of Cultivated Species and Geographic Distribution of Crop Diversity

Although the first land plants, the embryophytes, appeared some 515–470 million years ago in the Middle Cambrian–Early Ordovician period [20], humans who domesticated plants appeared much later, just 195,000–160,000 years ago and migrated from Africa about 130,000–120,000 years ago [21]. They were roaming the wild as hunter–gatherers, relying on wild plants and hunting wild animals for their food for another ~110,000 years before they settled to cultivate plants and rear animals in the Neolithic period—the start of the domestication of species (Table 1). Thanks to plant domestication and the resulting availability of food, humans became the predominant, most successful species on Earth, with a subsequent population explosion (Table 1) that demanded for more and more food.

### 2.1. Vavilov’s Centres of Origin of Cultivated Plants and the Theory of Homologous Series of Variation

The domestication of crops took place independently in eight geographic regions, described by Nikolai Vavilov [22] as primary centres of origin of cultivated plants in 1926 (Figure 1, Table 2). The criteria for these centres were high varietal diversity of the crop, presence of wild ancestors along with the domesticated ones, and a long history of the crop in the region. Vavilov was the first to observe that diversity of cultivated plants was not distributed equally around the world, but was associated with ancient civilisations. It was in the 1920s that he recognised the existence of these centres and started collecting germplasm through numerous expeditions [23,24]. He established the world’s first genebank in Petrograd (Saint Petersburg), with branches across the Soviet Union where almost any crop could be grown because of the vastness and diverse climatic zones in the country. The current status of this genebank and its activities are described by Dzyubenko [25].

During his study of cultivated plants and wild species, Vavilov noted regularities of genotypical and phenotypical variation within the polymorphism between related species of the same genus, between species in related plant genera of the same family, or even between close families [26]. The first law of homologous series of variation states that “*closely allied Linnean species are characterized by similar and parallel series of variation; and, as a rule, the nearer these Linneons are genetically, the more precise is the similarity of morphological and physiological variability. Genetically nearly related Linneons have consequently similar series of hereditary variation*”. As a sequence to the first law, the second law states that “*not only genetically closely related Linnean species, but also closely allied genera, display similarity in their series of phenotypical, as well as genotypical, variability*”. Thanks to this discovery, the variation within a less studied species can be predicted if the variability of a related species is known. Voigt [27] has illustrated practical examples of the use of such predictions by Vavilov in plant breeding.

Such parallelism has been observed in recent molecular analyses of related crop species, for example, in resequenced *Brassica rapa* and *B. oleracea* [28]. Understanding of the existence of such parallelism is important in plant breeding because if a variant is predicted to be present, the required genotype can be found through hybridisation and selection in segregating populations or using mutagenic techniques [29,30]. Use of induced mutations to produce semi-dwarf rice mutants in *Japonica* and *Javonica* backgrounds in California [31,32] and phytophthora (*Phytophthora nicotianae* var. *parasitica*)-resistant sesame (*Sesamum indicum*) mutants in Sri Lanka [33,34] are some examples of such practical use.

Study of crop domestication is complex, and was especially so during Vavilov’s time because of difficulty in accessing the many areas of crop domestication spanning different continents. Later studies have shown that high varietal diversity does not exist for some crops in the centre of origin [36]. Archaeological findings in recent times have added further complexity to the theme of crop domestication. This has revealed smaller independent areas of domestication within large centres. For example, Vavilov considered India as one major centre. Archaeological evidence suggests that there are five independent centres within India [37]. Other broad regions, such as Near Oceania, Amazonia, Eastern North America, and the river deltas of Western Africa, have also been identified as a result of new archaeological and molecular biological findings. Thus, for example, the Niger River basin in West Africa is now considered a major cradle of crop domestication in Africa, with sequencing and microsatellite marker studies confirming the origin of African rice (*Oryza glaberrima*) [38], African yam (*Diascorea rotundata*) [39], and pearl millet (*Pennisatum glaucum*) [40] in this area. Li [41] discussed the complexity of Vavilov’s Chinese Centre and identified four belts of origin of cultivated plants in this region: I. Northern China, II. Southern China, III. Southern Asia, and IV. Southern Islands, according to their latitude. Archaeological evidence also suggests that in the early period, the focus on domestication was on a few crop species, mainly cereals. Plant domestication resulted in a lifestyle change for humans, from foraging to a more sedentary lifestyle. Increased agricultural productivity supported larger populations, and the first civilisations arose as a result. Purugganan and Fuller [42] describe 24 regions of domestication, most of them in and around Vavilov’s main centres.

### 2.2. Landrace and Modern Cultivars; Their Genetic Structure in Relation to Diversity Management

In later periods, large-scale breeding programmes or natural adaptation of crops outside the main centres after introduction by humans resulted in the diversity of particular species that Vavilov called secondary centres of diversity. The main difference in these secondary centres is the poor representation of wild relatives. In primary centres where the crop was domesticated, one sees wild relatives including the progenitors. Genetic diversity studies of domesticated crops and their wild ancestors can provide insight into the history and timing of domestication, shedding light on the food habits of our ancestors. Additionally, we can understand the genes that underlie the main phenotypic and genetic shifts in populations leading to domestication events, giving clues for the better use of underutilised crops for breeding. Most importantly, from crop genetic diversity studies, we can identify genetic groups within populations that need to be retained as germplasm for conservation and utilisation. The value of wild populations is in the large genomic variation and novel genes and alleles they carry that can be introgressed into cultivated species where there is typically lower genetic diversity due to domestication and selective breeding [43,44,45]. Therefore, once candidate genes associated with adaptation to emerging biotic and abiotic stresses in wild populations are identified, they can be introgressed into new cultivars. This approach can contribute to increased resilience of the cultivated species in new crop varieties destined to keep feeding the increasing population under climate change.

Soybean (*Glycine max*) is a well-studied domesticated crop, arising in an area covering parts of present-day China (Manchuria), Korea, and Southern Russia, where its progenitor *G. soja* exists [46]. The genetic diversity measured using different genetic markers, such as simple sequence repeat (SSR) [47], single-nucleotide polymorphism (SNP), 5S ribosomal RNA polymorphism [48] markers, and a de novo assembly of sequence data [44] all point to the larger variation and presence of novel alleles in *G. soja* compared with the domesticated *G. max*, including rare alleles [43]. Furthermore, the genetic diversity of the cultivated species in the centre of origin is far greater than in secondary centres. For example, North America is a secondary centre of diversity of soybean, with more than 30 million ha and 2242 cultivars registered in the USA alone within the period 1970–2008 [49]. However, only a limited number of ancestral introductions have contributed to the germplasm developed there, with only five introductions being the cytoplasm source for 121 of the 136 cultivars studied by Specht and Williams [50]. The ancestry of nuclear material also was narrow, with only 12 introductions contributing to 88% of the germplasm [50]. This indicates the value and need for conservation of genetic resources at the source of origin, with special attention to wild progenitors and other wild relatives because of the presence of a wider diversity of alleles.

Despite the narrow genetic variability, secondary centres of diversity offer valuable agronomic traits in their germplasm; hence, some countries can directly adopt some of these cultivars until national breeding efforts commence. A good example was the adoption of U.S. soybean cultivars such as ‘Hardy’, ‘Lee’, ‘Improved Pelican’, ‘Davis’, and ‘Bragg’ in north-central India in the 1960s [51,52] and in Sri Lanka in 1974 [53] until local breeding programmes were initiated resulting in superior cultivars suitable for release [52,54].

Few cultivated species have been domesticated outside their region of origin. For example, sunflower from South America was developed into oilseed sunflower in Russia [55,56,57]; grapefruit, a hybrid of *Citrus sinensis* from Southeast Asia and *Citrus maxima* from Indonesia, was domesticated in Barbados in the 1820s [58]; and Chinese gooseberry (*Actinidia* spp.) from East Asia was domesticated and commercialised in New Zealand as kiwifruit [59,60]. Again, the diversity of wild species is richer in the source region, with many wild species of sunflower in South America, and 57 species of *Actinidia* in China [61] against 19 in New Zealand introduced before China embargoed further export of kiwifruit genetic resources [62].

## 3. Crop Domestication and Domestication Traits

### 3.1. Primary Domestication Traits

Humans who had traditionally foraged (Table 1) started cultivating limited plant species as food sources in the early Neolithic period (13,000 to 11,000 years ago). The morphological, physiological, and biochemical changes in species during evolution can take different directions under domestication depending on the part of the plant used. For example, in cereals and pulses, there is evidence for an increase in grain size and non-shattering at maturity [22,42]. As a result of acquiring the non-shattering grain character, cereal crops lost their ability for dispersal and became dependant on humans for reproduction by sowing. Many tuber and root crops, on the other hand, lost the ability to sexually reproduce as a result of selection for larger tuber or root, associated with selection for polyploid types resulting in sterility. In *Diascorea alata*, a dioecious tuber crop, all 73 male genotypes studied by Abraham and Gopinathan Nair [63] were tetraploid, whereas most of the 30 female germplasm accessions were of higher ploidy (hexa and octaploids) and completely sterile.

Meyer and Purugganan [64] describe domestication traits as those selected during the initial transformation and establishment of a new domesticated species from its wild ancestor. Changes in resource allocation to the part of the plant commonly used for food are typical in many crop categories, whereas loss of dormancy, determinate growth habit, increase in seed size with thinner seed coat, and changes in inflorescence architecture are hallmark domestication traits in seed crops. These traits arose either through conscious human selection or ability of the particular genotype to survive under deforested or disturbed habitats. Traits to facilitate harvest (e.g., non-shattering in cereals) represent the former, and larger seed size the latter because of the ability of larger seeds to emerge after burial during planting (competitive advantage) [37,42,65].

Archaeological evidence from wheat, barley, and rice suggests that the seed size increase was one of the first traits selected under cultivation followed by the non-shattering of grains, the latter taking a much longer time for fixation. Seed size increases in barley and rye were achieved within 500–1000 years, whereas in rice it happened at a much slower pace, over the period 9000–5500 years ago [42,66,67]. On the other hand, pearl millet seed size enlargement occurred only 2000 years after domestication [42] and occurred at several locations [66]. Similarly, a 1500–2000-year period for seed size enlargement in mung bean is evident from recent archaeological findings from Indian sites [65,66].

For the first time, Li et al. [68] cloned and characterised a gene, *sh4*, for the loss of shattering in a grain crop (in rice), a hallmark trait of domestication, which, interestingly, is expressed at a slower rate during grain maturation in cultivated *O. sativa* than in the seed shattering progenitor *O. nivara*. This was probably the result of selection in the regulatory region of the gene for finer adjustment of the shattering/threshing balance in cultivated rice.

Another trait of domestication is tillering and branching. Generally, in most seed crops, apical dominance has increased, with the suppression of lateral branching or tillering during domestication. Doebley et al. [69] cloned *teosinte branched I* (*tb1*), the key gene contributing to the increased apical dominance in maize (*Zea mays* ssp. *mays*) compared with its ancestor teosinte (*Z. mays* ssp. *parviglumis*) (Figure 2). Their research led to the discovery that *tb1* acts both to repress the growth of axillary organs and to enable the development of female inflorescences. The gene is not expressed in the primary axillary meristems of teosinte, enabling them to develop into long branches with a tassel at the tip (Figure 2). During domestication, the selection of forms that have high expression of *tb1* in primary axillary meristems led to the development of ear shoots rather than elongated tassel-tipped branches. Thus, during maize domestication for less branching, an alteration of the gene regulation of *tb1* has occurred rather than loss/gain or change in function. The most critical step in maize domestication was the liberation of the kernel from the hardened, protective casing that envelops the kernel in teosinte. This evolutionary step exposed the kernel on the surface of the ear, such that it could readily be used by humans as a food source (Figure 2). Wang et al. [70] mapped the factor controlling the phenotypic difference between maize and teosinte for this trait to a 1 kilobase region, within which maize and teosinte show only seven fixed differences in their DNA sequences. They demonstrated that this key event in maize domestication is thus controlled by a single gene (*teosinte glume architecture I* or *tga1*), belonging to the SBP-domain family 2 of transcriptional regulators [70].

Plant size can be reduced either as in the case of wheat and rice during the Green Revolution or from indeterminate growth habits to determinate growth, such as in beans and soybean. Plant architecture has been selected to suit harvesting practices, e.g., single-ear corn or single-head sunflower. Plants are also selected for ease of handling, thus losing their natural protective features such as thorns and spines in the case of some *Citrus* spp. and *Solanum* spp., respectively.

### 3.2. Diversification Events

After initial domestication, crops underwent diversification as a secondary event. For example, sticky and aromatic rice and popcorn were selected from the commonly grown rice and maize types, and a whole range of *Brassica oleraceae* vegetables (kohlrabi, cabbage, Brussels sprout, kale, broccoli, and cauliflower) were selected from mustard (Figure 3) as a result of diversification events. Most of the diversification traits evolve under targeted selection. Another interesting vegetable that has experienced through much diversification is lettuce (*Lactuca sativa* L.). Grown and used in ancient Egypt and depicted in Egyptian ancient art circa 2500 BC, it was a plant with narrow, erect leaves with prickles [71]. It has been used as an oil-yielding crop, and de Vries [72] considers that it was domesticated even earlier in the Kurdistan–Mesopotamia area. The fact remains that conscious selection has produced the many forms of lettuce in production today: cos, stalk, butterhead, crispbread (iceberg), and Latin. The oil type of lettuce is still used in Greece as a soporific [71].

Purugganan and Fuller [42] reviewed the genes directly involved in crop domestication that have been isolated and characterised to date. Of the nine domestication loci identified, eight encode transcriptional activators, including rice shattering genes *sh4* and *qSH1*, maize architecture gene *tb1* (suppresses axillary branch formation), and wheat inflorescence structure-determining *AP2*-like wheat gene *Q*. In contrast, more than half of the 26 genes involved in diversification where molecular function has been characterised encode enzymes. Thus, domestication events are associated with transcription regulatory networks, whereas crop diversification involves a larger proportion of enzyme-coding loci.

### 3.3. Physiological and Biochemical Changes

During domestication, changes in morphological features as well as physiological and biochemical features, such as photoperiodism, vernalisation requirements, and seed and tuber dormancy, were very common. Changes in life cycle to suit different seasons in different climates have occurred. Some crops have been turned into annuals from their original perennial habit, such as cotton, castor, pigeon pea and cassava; others have lost their natural protective toxins, e.g., many crops of the Solanaceae family, as humans selected against those features. Another example of major physiological change following selection is the evolution from wild *Ananas bracteatus* to domesticated pineapple *A. comosus* (Figure 4).

### 3.4. Genetic and Cytogenetic Changes

Polyploidy, the increase in genome copy number, is a central feature of plant diversification. This could be autopolyploidy, where the same genome is represented multiple times (whole genome duplication) as a result of sexual polyploidisation via unreduced gametes, or somatic polyploidisation followed by sexual reproduction [30]. A classic example is cultivated potato (*Solanum tuberosum*) (2*n* = 4× = 48), which has tetrasomic inheritance.

The other common form of polyploidy in domesticated species is allopolyploidy (amphidiploidy), where genomes of two or more species are represented in the new species. Durum wheat (*T. durum*) (2*n* = 4× = 28; tetraploid) and bread wheat (*T. aestivum*) (2*n* = 6× = 42 hexaploid) are well-known examples. Canola (*B. napus*) provides an example of how heterozygosity at polyploid level can increase selection advantage. It carries genomes of *B. oleraceae* (2*n* = 18, CC genome) and *B. rapa* (2*n* = 20, AA genome). Using quantitative trait loci analysis for yield, it was shown that canola yields were lower when it had allelic arrangements similar to the parental types, but when the arrangement differed from those of the parents, the yield was higher [73].

The majority of cultivated bananas are derived from inter- and intra-specific crosses between two diploids (2*n* = 2× = 22): *Musa acuminata* (AA genome) and *M. balbisiana* (BB genome). The parent species have seeded fruit with little starch and are of no value as a crop (Figure 5). Most of the cultivated bananas are parthenocarpic seedless triploids (2*n* = 3× = 33) with AAA (dessert bananas), AAB, or ABB (mostly cooking bananas) genomes. These variations have been collected from multiple independent sources in the wild; thus, the hybridisation events and mutations giving rise to the seedless and parthenocarpic characters have occurred many hundreds of times [74], meaning that bananas were domesticated in several areas of the Malayan centre of diversity.

## 4. Current Status of Plant Genetic Resources in the Centres of Diversity

As already described, along the path of domestication, humans have selected for only a few traits in our crops over time, resulting in the narrowing of the gene pool available for breeding. Landrace cultivars, although not as productive under high-input conditions of modern agriculture, carry alleles that are useful in many other ways, including pest and disease resistance, and tolerance to adverse soils, drought, salinity, and other abiotic stresses, while also carrying valuable quality traits such as a better nutritional value than many modern cultivars. For example, using whole-genome shotgun data from seeds of ancient and modern common bean, Trucchi et al. [75] showed that selection strategies during the past few centuries, as compared with historically, more intensively reduced genetic variation within cultivars and produced further improvements by focusing on a few plants carrying the traits of interest, at the cost of marked genetic erosion within Andean landraces. Using data from collecting missions and survey data since 1927, Hammer and Laghetti [76] found higher historical rates of genetic erosion of wheat in Italy (13.2%) compared with the period after 1950 (0.48–4%). Similar trends in genetic erosion have been found to occur in rice in India and China, *T. durum* and *T. dicoccum* in Ethiopia, and in traditional wheat varieties in Greece [77]. Crop wild relatives (CWRs) evolving in their natural habitat can also have many useful traits. Contemporary plant breeders are aware of the need to broaden the genetic structure of the crops they breed, although it is not easy to incorporate this aspect into breeding schemes because hybridisation with landrace cultivars, more so with wild species, requires time-consuming back-crossing to achieve the high yields and other traits expected under modern cultural practices. As a result, the gene pool of our crops often continues to narrow in breeders’ hands; at the same time, because of urbanisation, deforestation, monocropping, etc., the genetic diversity of crops is reducing at the centres of diversity.

Some of the centres of origin of cultivated species have recently become areas for large-scale irrigation and hydroelectric projects. By 2000, there were over 45,000 large dams in more than 150 countries, and each year 160 to 320 new schemes are being built worldwide [78], at the expense of habitat loss for terrestrial ecosystems. For example, the world’s largest dam, the Three Gorges Dam in central China, was inserted in the middle of a biodiversity hotspot. Located in the upper reaches of the Yangtze River, little affected by the quaternary glaciations, the Three Gorges Reservation Area is one of the richest areas in biodiversity in China and was considered to have had the highest diversity of genera and families globally. It is the home of 6388 species of higher plants, belonging to 238 families and 1508 genera, and accounting for 20% of all seed plant species in China, with 57 of them being endangered [79].

In addition to industrial development, in some centres of crop diversity, protracted wars have induced the displacement of farming communities, in some cases with the complete abandonment of farmland along with the crop genetic resources they contained. Good examples are in the Middle East—Syria, Iraq, Yemen, and Palestine. The depletion of crop genetic resources is thus happening both at the centres of diversity, in what is left of the natural environment, and on farmers’ lands. This is part of the larger problem of environmental degradation, and the COVID-19 pandemic could be viewed as a symptom of a bigger problem of deforestation and biodiversity loss that needs addressing urgently.

Without continued genetic enhancement using diverse germplasm from both CWRs and landrace cultivars, gains in crop yields obtained over the past seven decades are not sustainable, and yields might eventually grow more slowly or even decline, as already discussed [16]. Hence, comprehensive integrated programmes are needed for the conservation of plant genetic resources.

## 5. Role of Perennial Species in Sustainable Agriculture

Plants show a wide variability in the distribution of the limited resources available, allocating them to growth, defence, and reproduction, with two contrasting strategies: annual species, which complete their life cycle and die within a year; and perennial species, which usually delay their flowering to a later year, sometimes interrupted by periods of quiescence. Ensuring the need for agricultural products by a growing and more demanding world population through the intensification of conventional agriculture without causing significant damage to the environment is unrealistic. In recent decades, the environmental costs of intensifying conventional agriculture have begun to cause serious concern. Ecological intensification has been proposed as a nature-based alternative that complements or partially replaces external inputs [80]. Perennial plants are key components that offer previously unattainable levels of ecological intensification in agriculture, reducing the impact on the environment [81].

Although annual crops are the main source of our diet, several perennial crops could also be important players in the coming decades. Approximately one-eighth of the total area of global food production is composed of perennials, which are therefore a fundamental source of nutrition worldwide [82]. Perennial crops have several advantages from the point of view of environmental impact. One benefit is that they do not have to be reseeded or replanted every year, so they do not require annual ploughing to establish. Moreover, to successfully grow annuals, farmers must chemically or mechanically control weeds to avoid competition with crops for light, nutrients, and water, especially in the early stages of seedling development. The resulting soil disturbance has caused significant amounts of carbon loss in the soil (which ends up in the atmosphere as CO_2_), soil erosion, nutrient loss, and an impact on soil organisms [83]. From a study on the dynamics of soil organic carbon, it was estimated that over a 20-year period, encompassing a change from annual to perennial crops led to an average 20% increase in organic carbon at 0–30 cm soil profile [84]. Compared with organic and conventional cultivation systems of annual wheat, recently commercialised perennial intermediate wheatgrass (IWG) cultivation increased the soil organic carbon in 30–60 cm soil depth, including the amount of carbon in the particulate organic matter, implying reduced carbon losses and high carbon use efficiency [83]. Another study found that carbon flux, as well as carbon and nitrogen storage in soil were greatest in IWG systems compared with both restored native vegetation and the annual monocropping rotation of wheat, sorghum, and soybean [85].

In general, compared with annual crops, perennial crops are more robust, protect the soil from erosion and improve its structure, increase the retention of nutrients, organic carbon, water [83,84,85], and therefore can contribute to the adaptation and mitigation of climate change. Overall, they help ensure long-term food and water security. Another advantage of perennial crops is that they can free farmers from economic instability by significantly reducing tillage and planting costs and their time in the field. In recent years, research has contributed to improving agricultural techniques and practices to support environmentally friendly agricultural systems based on perennial crops. All these ecological benefits were recently proven in 14 woody perennial polyculture farmlands when compared with annual monocultures in the U.S. Midwest, one of the most industrialised food-producing regions in the world [86]. A similar beneficial effect was found in semi-arid West Africa when the correct woody perennials and shrubs were chosen for growing along with annual cereals or legumes, particularly in farmer-managed natural regeneration systems as an agroforestry strategy [87]. However, perennial species are particularly recalcitrant to conventional breeding programs, because there are many obstacles to their improvement when compared with annual crops [82]. Many perennial species have long juvenile phases and thus require up to several years before the yield and quality can be evaluated. Not only are the time, space, and infrastructure required for breeding perennials often far greater than for breeding annuals, but the evaluation of commercially relevant traits is also often more complex, time-consuming, and expensive [82].

### Herbaceous Perennial Crops

Agroecosystems are in constant evolution, in order to adapt to the needs of a growing population in a sustainable manner. Debates on the ecological impacts of agricultural intensification, including soil degradation and erosion, have concentrated attention on crops that provide both agricultural products and ecological services. Annual crops sown every year deplete the soil and expose it to erosion, requiring weed control with herbicides. Moreover, during the first phases of growth, shallow root systems are not efficient in absorbing water and nutrients, resulting in ground and surface water pollution by nitrate leaching [88]. Perennial herbaceous crops, which can be harvested mechanically, are perceived as a sustainable alternative to annual crops used as a source of human and animal food. They can grow for several years, produce a large root system that helps to reduce soil erosion and increases the soil organic matter, and can support several below-ground microbial communities that make plants more competitive, with better performance and, at the same time, with a lower impact on the environment. However, these species, which provide an opportunity to explore potential alternatives to annual crops, have been almost absent from agriculture, and were rarely domesticated for seed or fruit production [81,84].

Different possible biological mechanisms have been proposed to explain the rareness of herbaceous perennial crops. Among these, compromises between vegetative and reproductive tissues stand out, and it is now possible to develop these through phenotypic and genotypic selection [89]. The presence of genome sequences in wheat relatives such as *Thinopyrum intermedium* that may be orthologous to domestication genes identified in annual grain crops gives optimism [88]. Plants must devote limited resources towards a variety of processes, including growth, defence, and reproduction. In wild species, allocation strategies must reflect trade-offs between these processes and are therefore central to a species’ ecology. During the domestication process, artificial selection on allocation was carried out, which generated high-yielding crops that often invest reduced resources in defence or longevity, thus limiting the number of herbaceous perennial crops [89,90].

Need in developing new perennial crops and better understanding domesticated perennials has led to an increase in interest in the physiology, genetics, and evolution of these species. Our ability to understand or predict evolutionary transitions between strategies and their adaptive significance is limited by the lack of integration between the different fields. The evolutionary transition between perenniality and annuality in plants is exceptionally common among angiosperms. Reconstructions of the ancestral status using phylogenetic approaches have generally found that annuals derive from perennial ancestors; however, the evolution of perenniality was also observed. Empirical data support the hypothesis that evolutionary changes to annual life histories are associated with arid, disturbed/unstable environments in which adult survival is low or unpredictable [91].

The interest in developing new perennial crops through wide hybridization led to the crossing of annual crops with perennial relatives and the de novo domestication of wild, perennial, herbaceous species [92]. However, the idea of developing new perennial crops to replace annual grains is controversial [89,93]. The opportunity costs associated with the low grain yield compared with the high yield index of annual crops are one of the most persistent criticisms of perennial crops [94]. In his review, Smaje [94] raises three arguments against developing perennial grain agriculture: (a) ecological theory suggests that perennial grains may yield less than annual grains; (b) strong criticisms of annual agriculture are unfounded, both socially and ecologically; and (c) focus on perennial grains detracts from more important strategies for achieving agricultural sustainability. Crews and DeHaan [95] counter these three arguments, concluding that perennial herbaceous crops constitute a valid solution for enhancing sustainability in agriculture.

Many researchers have suggested that a more sustainable agricultural system will need to consist of mostly perennial species, and will need to be more diverse than is the case with present agroecosystems [96,97]. In the early 1980s, Wes Jackson [98] proposed the idea of growing perennial grain crops, including cereals, pulses, and oilseeds, and by planting them in complementary arrangements in prairie. This idea, which was considered utopian by many researchers in the past, seems to be achievable today thanks to new plant biotechnologies. In fact, the rapid deployment of technologies, such as CRISPR-Cas systems and other genome editing processes, provides new opportunities for crop breeding. The de novo domestication strategy can be adopted to improve the elite foundation materials from wild or semi-wild plant species in nature to achieve the main goals of genetic improvement, followed by the introduction of desired traits through the application of new genetic engineering technologies while retaining their desired features, resulting in plants harbouring new traits of interest [99]. If the new technologies that have been developed in recent years are applied to perennial proto-crops, it would enable the development of new genotypes of interest, probably in decades as opposed to the centuries that were required to create our current annual food crops [96].

The idea of rapid neodomestication was propagated as a promising strategy for future sustainable agriculture [100]. Despite accelerating the process, neodomestication through crosses would still suffer from the accumulation of deleterious mutations linked with domestication traits alleles, the so-called domestication cost [101]. Although the scientific community is not completely unanimous on the potential impact of crop neodomestication, there are several successful studies rendering this concept of potential interest to agriculture [102]. Neodomestication was first utilized for a rapid sunflower breeding program that introduced domestication traits into wild relatives through crosses [103]. In this regard, the high phenotypic gains, approaching 320% in breeding sunflower as a perennial oil crop [104], are encouraging. Subsequently, species of the genus *Vigna* were nominated as candidates for neodomestication due to their stress resilience and common use as edible wild plants [105].

Identifying the right herbaceous perennial species and germplasm for domestication can be challenging. The construction of an online resource of wild, perennial, herbaceous species—the Perennial Agriculture Project Global Inventory (PAPGI), comprising details of perennial members of the three of the largest known plant families, Asteraceae, Fabaceae, and Poacea, containing details of taxonomy, growth descriptors, ecology, reproductive biology, genetics, economic uses, and toxicity [106] in this regard is invaluable. As the first component of the PAPGI project, focus has been on wild, perennial, herbaceous Fabaceae species, with records of 6644 species and over 60 agriculturally important traits. Food and forage uses of 314 legume species and toxicological data for 278 species have been incorporated into searchable online resources [81].

## 6. Approaches to Germplasm Conservation

There are two main approaches to conserving crop germplasm: in situ and ex situ. Article 2 of the Convention on Biological Diversity defines ex situ conservation as the conservation of components of biological diversity outside their natural habitat. In situ conservation relates to the conservation of ecosystems and natural habitats, and the maintenance and recovery of viable populations of species and, in the case of domesticated or cultivated species, in the surroundings where they have developed their distinctive properties. Thus, in situ conservation has two major facets: (a) the conservation of ecosystems and natural habitats, mainly facilitating the conservation of CWR in the natural ecosystems; and (b) the conservation of domesticated species in the habitats where they were developed [107]. The latter is called on-farm conservation and is part of wider in situ conservation.

### 6.1. In Situ Conservation

In situ conservation is important for conserving CWRs and landrace varieties of the cultivated species. In 1989, the Commission on Genetic Resources for Food and Agriculture (CGRFA) called for the establishment of networks of in situ conservation areas for plant genetic resources for agriculture, for both crops and CWR.

#### 6.1.1. Identity of Crop Wild Relatives

CWRs are commonly defined as wild species that are relatively closely related to agricultural and horticultural crops. Therefore, any taxon belonging to the same genus as a crop would fall into the CWR category. However, this definition covers large numbers of taxa and can result in the inclusion of species that are too remotely related to the crop or a large proportion of the species, e.g., in the Mediterranean Region and Europe almost 80% of flowering species are CWRs [108]. Considering the limited resources available for ex situ conservation efforts, a more rational approach would be to use the gene pool concept [109], where the close relatives are categorised as the primary gene pool, the more remote relatives as the secondary gene pool, and the most distant relatives as the tertiary gene pool. For many tropical species, where the relatedness in terms of crossing ability and genetic diversity is not well understood, the taxonomic hierarchy may be used to identify the relatedness of CWR to the cultivated species [110]. Even if relatedness is not yet established, new-found taxa can become high priority for conservation, such as the case with *Oryza rhizomatis* Vaughan, discovered in the late 1980s, mainly occurring in the driest areas of Sri Lanka (Figure 6). It is rhizomatous [111] and has drought-avoidance traits. It is in the near-threatened category in the IUCN Red List [112] and is a high priority rice taxon in terms of in situ conservation with another three species (*O. longiglumis*, *O. minuta*, and *O. schlechteri*).

#### 6.1.2. Why In Situ Conservation?

The evolution of crop species is continuing in the centres of diversity of crop plants, with new landraces emerging and genetic frequencies changing, as shown, for example, in the evolution of cultivated rice [114]. Thus, the main difference in maintaining the diversity in situ as against ex situ conservation in field genebanks or seed banks is that we have a continually evolving population responding to the changing environment. Allele frequencies vary over time in response to the changes in environment, thus making available genotypes of particular interest for contemporary plant breeding problems. Older ex situ collections, if not updated, remain frozen snapshots of a particular epoch of evolution and will subsequently have genotypes that are not adapted to the changed environment of the original collection site.

In addition to the conservation of genetic resources in the wild, on-farm conservation also helps to protect the conditions that allow the emergence of new germplasm. This idea of dynamic conservation extends to the whole farming system. This type of conservation enables the maintenance of genetic resources at all levels—ecosystem, species, and intraspecific—supporting and contributing to the overall agroecosystem health in locally tested farming systems. This includes minimising the use of pesticides, restricting emissions, conserving soil, and preventing pest and disease outbreaks as multiple crops and heterogeneous varieties within a farming community provide less than ideal conditions for such outbreaks.

CWRs and landrace varieties provide new sources of variation for crop improvement programmes. There are many examples of novel cultivars produced using these genetic resources from the past and present. For example, when late blight (*Phytophthora infestans*) decimated the potato industry in Europe as a result of the import of infested seed potatoes from the United States in 1845, the introgression of phytophthora disease resistance from wild relatives, such as *Solanum demissum* from South America [115], helped to revive the industry; most modern potato cultivars carry resistance genes from this wild species. Potato blight caused one million deaths and the displacement of another one million people from Ireland, a country that was totally dependent on potato at the time [115]; a historical lesson of the importance of diversification of food sources.

In recent times, rice improvement has greatly benefitted from wild relatives. Brown plant hopper (BPH—*Nilaparvata lugens* Stål.) is one of the most destructive pests of rice throughout Asia, causing severe yield reduction by directly sucking the plant sap and acting as a vector of virus diseases such as rice grassy stunt and ragged stunt. *O. glaberrima* and *O. minuta* have durable resistance to this pest. Using embryo rescue techniques, this trait was transferred to cultivated rice [116]. The resulting lines are being used worldwide in rice breeding programmes. For example, screening of the introgression lines from the crosses IR 64 x *O. glaberrima* and IR 31917-45-3-2 x *O. minuta* showed that the two wild rice parents and the introgression lines had greater resistance to BPH than any of the local tolerant genotypes or IR 64, which has a *Bph1* gene which imparts some tolerance to BPH [117]. Examples of different wild rice species used in the improvement of targeted traits in cultivated rice are given in Table 3. Considering the value of such germplasm, both the Convention on Biological Diversity and the International Treaty on Plant Genetic Resources for Food and Agriculture (ITPGRFA—termed Plant Treaty) highly recommend the implementation of in situ conservation strategies.

#### 6.1.3. Implementation of In Situ Conservation

In general, implementing in situ conservation has been more challenging than ex situ conservation for several reasons. The in situ conservation of traditional cultivars is not well funded, unlike traditional genebank activities. Management and coordination activities of in situ collections have logistical problems because on-farm programmes have a poor connection with mainstream genebanking activities at both national and international level. In many environments, in situ conservation is still in an experimental stage and not supported through mainstream funding which is available for genebanking.

About forty years ago, it was widely assumed that traditional varieties would be rapidly and completely replaced by modern varieties [130]; this did not happen in several agricultural regions around the world [131]. For example, maize landraces in the U.S. corn belt were completely replaced between 1925 and 1950, whereas those in Mesoamerica continue to be cultivated [132]. A substantial amount of information has been documented in the last few years on the continuing maintenance and use of traditional varieties by small-scale farmers around the world [131,132,133]. Farmers seem to find that diversity, in the form of traditional varieties, remains important for their production systems. In fact, traditional varieties seem to adapt better than modern varieties to climate change and require lower chemical inputs. There are different ways of supporting farmers and farming communities in the maintenance of traditional varieties and crop genetic diversity within their production systems: (i) on-farm diversity assessment; (ii) access to diversity and information; (iii) the extent of use of available materials and information; and (iv) benefits derived by the farmer or farming community from their use of local crop diversity [131,133].

Different studies carried out around the world have demonstrated the value of the use of traditional varieties by small-scale farmers for conservation [131,134,135]. However, most studies on this topic suggest that there is insufficient knowledge about the social, cultural, and methodological dimensions on the topic, particularly how seed exchange networks cope under climate change, and under changes in socioeconomic factors, and family structures that have supported seed exchange systems to date [136,137]. Four core criteria have recently been proposed that characterize diverse Seed Commons arrangements at local and regional scales: (1) collective responsibility; (2) protection from private enclosure; (3) collective, polycentric management; and (4) the sharing of formal and practical knowledge [138].

A successful in situ on-farm conservation programme would usually have awareness raising as its first step. This will encourage not only the growing of local crops, but also their use by consumers. Local policy makers, journalists, and rural leaders, including farmers themselves, need to be educated through different activities, such as school programmes, poetry, essay and drama, village fairs, news, social media, etc. Through their daily interactions with the on-farm crop diversity and with neighbouring farmers, a local farming community is likely to know more about the local crop genetic resources than anyone else. This is a good reason for the incorporation of farmers into the national plant genetic resources (PGRs) system, making them partners. The interaction of genebank operators with the local community helps make farmers aware of their activities. This interaction will benefit both parties because it facilitates farmer access to genebank material, and the genebank will receive farmer cooperation to maintain crop genetic diversity in situ. Genebanks can also facilitate the communication between farmer groups scattered throughout a country, sharing resources and learning from one another. In this model, on-farm conservation recognises local farmers as the curators of the crop genetic diversity and links it with indigenous knowledge.

On-farm conservation should be targeted at uplifting the livelihoods of resource-poor farmers. This can be achieved if development efforts are targeted at local resources to empower farmers, leading to the sustainable development of their livelihoods. This can be approached through infrastructure development, securing new marketing opportunities for underutilised crops and varieties with identified nutritional value or other traits with consumer preference. With demand growing for organic foods, organic certification programmes can be introduced at village level. In many developing countries, where industrialisation/mechanisation has yet to occur in some areas and farming traditions go back many centuries, farmers have developed a sense of community and collaborative relationships where they exchange seeds, planting material, and labour. In contrast to this, in industrialised countries, farmers compete with each other and decisions are made based on economic reasons, with tradition having much less influence on how farming is practiced—this makes it more difficult to introduce on-farm conservation. Increased mechanisation has also seen farm sizes increase and mono-cropping replacing traditional farming systems. As a result of the introduction of modern agricultural practices in Germany, for example, 90% of the original diversity of landraces has been lost [139]. Nevertheless, with a strong genebanking tradition, Germany has managed to introduce on-farm conservation practices better than many other industrialised countries, with about 50 initiatives launched for in situ conservation [139].

Some wild species are threatened by overexploitation by communities living in the periphery of conservation areas. The domestication of such species and propagation through seed gardens with the participation of users has been proposed as an approach for the conservation of genetic diversity of such species [140]. Studies on the changes in the diversity of landraces on farms have been critically analysed and recommendations for further studies and conservation measures needed have been proposed in a recent review by Khoury et al. [132].

### 6.2. Ex Situ Conservation

#### 6.2.1. Origin of Genebanks and Their Spread

Nicolai Vavilov was one of the first to realise that the traditional crop varieties and land races were being lost from farmers’ fields where they originated. This led him to establish the genebank in Petrograd in the 1920s, with 50,000 seed samples collected from more than 50 countries as a result of his expeditions. Since then, several genebanks have been established, and seed exchanges with Western Europe, the USA, Australia, and New Zealand have commenced. Those that are large or with a particularly wide scope include the All Union Institute for Plant Industry (VIR), Leningrad (Saint Petersburg), established in 1920; the Empire Potato Collection, Cambridge, UK (1938), now operating from James Hutton Institute, Invergowrie, Scotland; the Rockefeller Foundation Collections of maize under the Mexican Agricultural Programme (1943) [141]; and the National Seed Storage Laboratory, Fort Collins, CO, USA (1958). Similar collection efforts were commenced in many other South American countries in collaboration with American Land Grant Universities. By 1952, the USA had established four plant introduction stations in Ames (Iowa), Pullman (Washington), Geneva (New York), and Griffin (Georgia). In Europe, other than the VIR in the Soviet Union, significant work was conducted at the Institut für Kulturpflanzenforschung in Vienna, where Hans Stubbe carried out collection missions: this institute was moved to Gatersleben in 1946 and now operates as the Leibniz-Institut fur Pflanzengenetik und Kulturpflanzenforschung (IPK).

#### 6.2.2. International and National Genebanks

The FAO, despite commencing the Plant Introduction Newsletter as far back as 1957, did not have any on-ground programmes for collection and conservation until 1964. This started to be addressed following an FAO Technical Meeting on Plant Exploration and Introduction, where the recommendation for setting up national and regional plant introduction centres was proposed and adopted. The FAO Expert Panel on Plant Exploration and Introduction was established in 1965 and held six meetings up to 1974, when the International Board for Plant Genetic Resources (IBPGR) was established. By the 1970s, many initiatives on international collaboration were in place, which gained further momentum with the establishment of the IBPGR [142] which, in 1991, became the International Plant Genetic Resources Institute—IPGRI. Within two decades, its network supported collections of over 200,000 accessions in 136 countries and coordinated the creation of an internationally linked system of genebanks called the Registry of Base Collections (Figure 7). Its aim is to conserve and make a subset of those materials available for national programmes [143]. The international genebanks were established in the centres of diversity of particular crops, but circumstances required moving some. For example, the war in Syria resulted in the relocation of the International Centre for Agricultural Research in the Dry Areas (ICARDA) from Aleppo, Syria, to Beirut, Lebanon, with most of the research activities moved to Morocco [144]. In addition to the traditional eight centres (Figure 7), Bioversity International holds a banana collection and supports aroid and yam genebanks in the Pacific. The Centre for International Forestry Research (CIFOR) and the World Agroforestry Institute have tree and fruit crop collections, and the International Livestock Research Institute (ILRI) has a fodder crops collection.

In 2006, IPGRI centres signed an agreement with the Governing Body of the ITPGRFA, as a result of which the work of the centres is now influenced by Plant-Treaty-related activities. In the 1970s and 1980s, many national genebanks were set up both in industrialised and developing countries. Generally poorer in crop genetic diversity, industrialised countries gave technical support for setting up national genebanks in developing tropical countries with a rich diversity of plant genetic resources. For example, Sri Lanka’s Plant Genetic Resources Centre was set up with the Japanese Technical Cooperation in 1988. The FAO estimates that 1750 genebanks exist worldwide with a total holding of about 7.4 million accessions. Of these, about 6.6 million are held in the national genebanks of individual countries [145]. For example, India’s National Genebank holds around 0.39 million accessions, with similar numbers spread across 41 National Active Germplasm Sites [146].

To further safeguard the collections, the Svalbard Global Seed Vault (SGSV) was opened in 2008 under a partnership between the Ministry of Agriculture and Food of the Government of Norway, the Nordic Genetic Resource Centre (NordGen), and the Crop Trust. It is a backup facility for all the genebanks around the world and holds 1.15 million accessions in some 5000 species. It has a capacity for 4.5 million accessions and seeds are held under black box conditions, i.e., the seed boxes and containers stored in SGSV will not be opened. The seeds are indisputably owned by the depositing genebank, and only that genebank can request the return of seeds stored in SGSV [147]. For example, when the access to ICARDA genebank in Syria was finally closed in September 2015 due to war, seed boxes safely deposited in SGSV were systematically retrieved, regenerated, and multiplied in Lebanon and Morocco for the continuation of ICARDA operations [144]. Thus, located halfway between Norway and the North Pole, carved into ice in the permafrost 110 m above sea level, SGSV provides back-up for individual collections in the event that the original samples in conventional genebanks are lost due to natural disasters, human conflict, changing policies, mismanagement, or any other circumstances [147]. Engels and Ebert [148] recently critically reviewed the current global system of ex situ collections in the context of political and legal frameworks.

### 6.3. Management of Ex Situ Collections

#### 6.3.1. Management in Time

Under ex situ conservation, once the plant or seed samples are removed from the centre where they have evolved, natural processes of selection and adaptation to the environment cease. Thus, the collected sample is a “frozen snapshot” of the genetic structure at the time of collection [149]. In the case of seeds, particularly in cross-pollinated species, the representativeness of the sample is further reduced every time it is regenerated, because of genetic drift and natural selective pressures under different environmental conditions.

In the centres of diversity, on the other hand, crop evolution is an ongoing phenomenon: the “loss” and origin of “new” alleles frequently occur; indeed, land race cultivar dynamics are quite high [149,150]. Even in extensive collections from the Mediterranean region, researchers have found low geographic coverage and poor representation of on-farm genetic diversity in ex situ collections established in Europe [151]. This is perhaps because as for wild species diversity, a stratified sampling strategy is required for full coverage [152]. Notwithstanding, no matter how well sampled, ex situ collections do not represent the natural population in the diversity hotspot, after a few decades. Therefore, a regular and systematic monitoring programme of the landrace cultivar population dynamics in well-defined in situ hotspots is needed to better understand the drivers of change [150] and to supplement existing collections. Systematic monitoring programmes would enable proper sampling at regular intervals to capture the changing allelic frequencies, which is particularly important for meeting breeding challenges in a changing climate.

#### 6.3.2. Identification of Duplicates

With large numbers of accessions in genebanks, another issue confronted by curators is the duplication of samples resulting from poor passport data and collection strategies, inconsistent documentation, and a lack of characterisation, amongst other factors. Labelling errors, hybridisation between stocks, and confusion about origins have also been identified as problems in ex situ collections. With over 7 million existing accessions in genebanks around the world [145] and increasing storage demands and costs, methods for efficient characterization and curation are required to avoid duplication. Genebanks have used molecular markers, such as simple sequence repeat (SSR), single-nucleotide polymorphism (SNP), amplified fragment length polymorphic (AFLP), and chloroplast DNA markers to characterise collections for their diversity and to identify duplicates with the aim of rationalising future collection and conservation efforts.

Lately, genome sequencing is being used to identify new patterns of variation [153,154], alleles of interest for breeding programmes [155], as well as duplicates [156] in collections. Methods are being developed to sequence large populations at low cost, including complex polyploid genomes [157,158,159]. Additionally, genome sequencing enables the unravelling of patterns of evolution of ancient crops, such as apple [160], grape [161], rice [114], and wheat [162], giving insights to their progenitors and conservation needs. Furthermore, they enable quantitative trait mapping and novel allele mining from large genetic collections [163]. When applied across different genebanks, these methods will enable the cost-effective and efficient management of germplasm and better stewardship of valuable genetic resources.

#### 6.3.3. What to Conserve and Use—The Concept of Core Collections

It is practically impossible to cover the entire range of landraces or the diversity within CWR in a genebank. To manage the increasing number of accessions in collections and the resultant difficulties in monitoring, regeneration, evaluation, etc., the concept of core collections arose [164]. Identifying ‘representative’ samples within large collections helps to better utilise genetic resources when there are tens of thousands of accessions to choose from with just basic passport data. Such subsets in a collection represent the maximum diversity without redundancy. This allows the supply of a set of accessions for evaluation or breeding purposes without compromising the diversity within the collection.

With the introduction of the core set concept, genebanks now has a tool to manage their collections more efficiently. Core collections have since become focal points for conservation prioritisation, phenotypic evaluation, genotyping, and exchange. In the early years, the objective was to develop a single entry point for users, providing them with the widest diversity within a manageable number of accessions. Collections were defined hierarchically using taxonomic characterisation, genomic distance, and geographic data, dividing the collection into clusters and selecting ‘core’ genotypes within those clusters using methods such as proportional allocation, log frequency allocation, and constant allocation groups. Once a core collection is developed, it can be validated using several methods, such as mean comparisons with the entire collection, the homogeneity of variances and frequency distributions among traits between the entire collection and the core, and optimising correlations [165].

Alternatively, diversity can be assessed using molecular markers, and core sets developed using a maximization of the number of alleles observed in each marker locus without relying on a stratification strategy. This M (maximization) strategy examines all the accessions for the alleles in the collection and identifies those that maximize the number of observed alleles at the marker loci. These can then be chosen as final candidates for the core. The superiority of this marker-based method is derived from the correlation between observed allelic richness at the marker loci and allelic richness at other loci. The software uses iterative procedures to select samples with the highest diversity as measured by the number of alleles and the trait classes that account for the greatest proportion of the collection variability based on the M strategy. The method can also be applied for both quantitative and qualitative data [166]. For example, this method enabled the production of a mini core subset of 217 accessions representing a core of 1794 accessions from the United States Department of Agriculture Agricultural Research Service (USDA-ARS) collection of rice, comprising more than 18,000 accessions [167], and to produce a core of 20 accessions within 450 apple accessions of the New Zealand apple germplasm repository, which was targeted for the first round of cryopreservation using winter dormant buds [168,169].

Core collections can also be developed using a multivariate distance approach [170] to develop the clusters followed by selection within those. Applying these methods, the International Crops Research Institute for the Semi-Arid Tropics (ICRISAT) has developed core sets from global collections of eleven crop species in ICRISAT genebank accessions [171]. Subsequently, mini core collections have been developed within the core collection using the same principles.

Core subsets for global collections, such as those maintained by the CGIAR, often serve as reference panels so that researchers around the world can evaluate the same genetic resources in different environments. Individuals in those reference sets may be selected as “controls” for phenotyping efforts (particularly relevant for disease and pathogen resistance/susceptibility and adaptation to different agro-ecological zones) so that results can be compared among research and breeding programmes. Some plant collections, particularly those that are clonally propagated, are difficult to distribute across international borders. Hence, multinational plantings of international core subsets for clonally propagated collections would help ensure that international communities have access to the same plant materials for comparative research.

### 6.4. Types of Collections in Genebanks, Their Management and Utilization

Ex situ collections are conserved in different forms depending on the reproductive behaviour, genetic composition of the population, use of the crop (seed, fruit, fibre, vegetative materials, etc.) and other considerations. These aspects are briefly discussed below, with references to recent studies for details.

#### 6.4.1. DNA Banks

Plant DNA banks were initially developed to create genetic libraries for evolutionary studies, to understand biological diversity, and to collect genomic information [172]. However, with habitat loss, species extinction is happening at a rapid rate, particularly in areas with high genetic diversity. Therefore, DNA banks are increasingly used to store genomic and diversity information of species in these vulnerable hot spots. Additionally, our ability to extract DNA from fossilised plant remains is providing new research opportunities in paleoecology, phylogeography, and evolution, including crop domestication. Therefore, the number of DNA specimens from extinct species is increasing in DNA banks.

Genomic research has seen unprecedented advances in recent years. However, the physical DNA from the published sequences is generally not accessible to researchers. Access to the original samples is important for conducting new studies, to extend or complement existing results, and to support good scientific practice, enabling the verification of published results. To address these needs, the Global Genome Biodiversity Network (GGBN) was formed in 2011, with the aim of developing high-quality, well-documented, and vouchered collections that store DNA or tissue samples, as well as to encourage and enable scientists to complete documentation chains between vouchers, tissues, physical DNA, sequences, and publications [173,174]. The GGBN currently has 99 member organisations from 35 countries, with over 4.1 million samples from 5152 families, covering 38,074 genera and 140,182 species. Their updates can be viewed in annual newsletters [175] and the data can be accessed through their data portal [176].

DNA banking is considered the most economical way of conserving genetic information of plant genetic resources, and is also the easiest way of exchanging genetic information across borders because DNA samples occupy less space, are more stable, and phytosanitary certification is not needed.

#### 6.4.2. Orthodox Seeds

There are three types of seeds from a conservation perspective. Orthodox seeds can be dried to low moisture contents (about 3% for oily seeds and about 7% for starchy seeds) without damage and be stored dry at low temperatures without losing their viability over long periods. Recalcitrant seeds cannot withstand desiccation to moisture contents below 20%. Intermediate seeds can be dried to a moisture content of 10% to 12%, but further desiccation reduces viability and/or dry seeds are injured by low temperatures. Seeds of most of the main crop species belong to the orthodox category, meaning that they can be safely cooled to standard long-term storage conditions of −18 °C without losing viability once dried to appropriate moisture contents. Seed storage is well researched and is the most efficient and cost-effective method of the conservation of plant germplasm, with about 96% of accessions held in genebanks worldwide as seeds [177]. In most of the genebanks, the seed vaults are maintained at −18 °C for long-term storage, whereas active collections are maintained at 5 °C to 10 °C. Seeds may be stored in dedicated cold rooms (Figure 8) or in domestic deep chest freezers or refrigerators, for which genebank standards and technical guidelines have been published by the FAO [178].

There are many advantages of conserving germplasm as seeds, including better security than in the natural environment, less space required, methods being simple, easily accessed and exchanged, and long storage periods. Additionally, seed collections capture more allelic diversity of the population than clonal collections. Therefore, in the case of fruit crop wild relatives, seed conservation is an option. However, as mentioned above, the evolutionary processes are frozen in time: as in any ex situ conservation method, regeneration is required from time to time. There can be gaps in the collection and initial set-up, and ongoing maintenance can be costly, including providing a constant power supply.

Under the ITPGRFA multilateral system, there are over 730,000 accessions available from the collections of the CGIAR system. The majority of these accessions are held in the form of seeds, with only 23,862 conserved as clones in vitro and 29,122 in field collections [143]. The number of accessions according to the species and the 11 participating genebanks are listed [143]. Additionally, the IBPGR has also coordinated the creation of an internationally linked system of genebanks known as the Registry of Base Collections (RBC) to conserve and make a subset of those materials available. Under this system, 144,000 accessions are available in 52 selected genebanks spread across all continents, covering 80 genera and 250 species [179]. Engels and Ebert [177] recently critically reviewed the current status of global seed banking with recommendations for improvement, emphasising the role of functional genomics and phenomics as well as strengths and weaknesses within regulatory frameworks and the strategies for linking national programmes with the global network.

#### 6.4.3. Genebanking of Clonal Crops and Recalcitrant Species

##### Field Collections

The traditional approach for maintaining germplasm of clonal species, species that are sterile or semi-sterile, and those producing recalcitrant seeds, is in field genebanks. They provide an excellent opportunity for curators to assess the diversity by phenotyping and for plant breeders to directly use accessions in their breeding programmes. Woody perennial crops (WPCs) represent about one-half of crop plant diversity and one-third of the 167 major crops. WPCs are usually clonally propagated because they are obligately outcrossing. Therefore, the conservation of germplasm through seed banks is not applicable to maintain the genetic characteristics of different heterozygous individuals. Moreover, the seeds of many species are recalcitrant, and therefore cannot be stored in traditional seed banks. Additionally, WPC species have a juvenile stage that can last for several years. In fact, plants generated from seeds usually exhibit strong, undesirable juvenile characters, such as a thorny habit and delayed flowering and fruit production [180]. In order to avoid these problems, the long-term conservation of WPC germplasm can be effectuated as field collections with clonal material in order to maintain the elite genotypes that form the foundation of woody perennial agriculture. However, WPCs represents only 5.8% of ex situ germplasm collections. Despite their importance, field collections are expensive to establish, and maintenance requires high costs for specialized technical personnel and land. Usually, woody perennials have a large plant size and therefore need large areas for maintenance in the field. Moreover, field collections have high risks of loss because they are exposed to natural disasters and are subject to biotic and abiotic stresses. In particular, the risk of pathogen infections transmitted through vegetative propagation is high and difficult to avoid. There are many examples of such cases, e.g., the loss of accessions in the apple germplasm collection in USDA due to fireblight (*Erwinia amylovora*) [181] and the loss of kiwifruit germplasm in New Zealand due to the incursion of *Psuedomonas syringae* pv. *actinidiae* [182]. Therefore, many genebanks back up their field collections in separate locations and also, more reliably, using lab-based conservation methodologies such as in vitro storage or cryostorage.

Establishing field genebanks may seem straightforward, but there are established best practices for sampling, the duplication of collections, and cataloguing the accessions. These can be found, for example, in the IPGRI training manual on “Establishment and management of field genebanks” [183], which has separate chapters on principles, legal issues, plant health, choice of materials, genetic considerations, planting layout, management characterisation and evaluation, utilisation, and economics. Another useful document is the IPGRI handbook No 7 on “Technical guidelines for the management of field and in vitro genebanks” [184]. Field genebanks of horticultural crops are a long-term commitment. Therefore, in many countries, field collections and genebank operations are undertaken by government departments.

##### In Vitro Collections

In general, lab-based germplasm conservation strategies fall under two major categories: slow growth procedures and cryopreservation. Both strategies require efficient regeneration system with high regeneration efficiency via organogenesis and/or somatic embryogenesis. In vitro storage has the advantage of maintaining collections under disease-free conditions and the cultures have fewer biosecurity requirements when material is exchanged across borders. On the other hand, because collections have high genetic diversity, the response to tissue culture media will be variable, which therefore requires prior research to optimise media for different species and even genotypes within species. There have been many studies on the tissue culture of cultivated species; therefore, this is not as daunting as it used to be a few decades ago. In vitro genebanks enable the rapid multiplication of material when required and provide a safe environment for managing germplasm collections in a confined space away from the field. To avoid somaclonal variation (genetic variation induced under tissue culture conditions), pathways using dedifferentiation and adventitious regeneration should be avoided. Therefore, intact shoot tips and axillary buds are typically used with minimal use of plant growth regulators.

After initiation and the multiplication of accessions in tissue culture, they are stored under slow-growth conditions to increase the period between subcultures, thus significantly reducing the cost of labour and materials. Generally, this is achieved by a combination of several factors: (a) decreased light (generally about 5–10% of the standard culture conditions); (b) low temperature between 4 and 21 °C with tropical species requiring higher temperatures; (c) lower concentration of mineral nutrients and sucrose in media, often without growth regulators; (d) smaller size of culture vessel/vial; (e) inclusion of growth retardants in media; and (f) osmotic stress using chemicals [62,185,186]. Unless available in the literature, these conditions need to be determined for each species by experimentation. Many genebanks around the world have in vitro storage for some species. Genebanks holding large numbers of accessions in vitro are given in Table 4.

Synthetic seed technology (SST) is an innovative and sustainable approach to preserve the biodiversity of clonally propagated woody perennials. A synthetic seed is defined as an artificially encapsulated somatic embryo, vegetative bud, or any other micropropagule that plays the role of a seed and has the ability to give rise to a complete true-to-type plant. SST offers opportunities to conserve clonal genetic resources safely at low cost.

General guidelines for the storage [187] and status of in vitro storage under the CGIAR network, particularly CIP, IITA, CIAT, and Bioversity International [188], have been published. Additionally, in vitro techniques have proven useful in collecting germplasm when seeds are not available (off-season), or in situations where seeds are not likely to remain viable because of their recalcitrant nature. Pence et al. [189] recently reviewed basic approaches and principles of in vitro collection, and one of the crop species for which in vitro collection is routinely used is coconut.

##### Cryopreserved Collections—Stopping the Biological Clock

Cryopreservation is the storage of biological samples in liquid nitrogen (LN) at −196 °C or in its vapour phase (LNV) at −165 °C to −196 °C. The demonstration that winter shoots of woody species can be conserved in liquid nitrogen in 1960 by Sakai [197] gave the impetus to undertake research on cryopreservation of fruit tree germplasm [198]. In the meantime, Latta [199] demonstrated the survival of carrot and sweet potato cell cultures in LN when pre-treated with high concentrations of sucrose followed by freezing to −40 °C and subsequent transfer to LN. This method came to be called classical two-step freezing or slow freezing. However, it was the advent of vitrification methods in the 1990s that allowed genebanks to take up cryopreservation on a mass scale as it became applicable across different families and genera. The method relies on the dehydration of cellular content to the extent that sudden freezing does not allow water molecules to form lethal ice crystals, but enters a state of metastable glass—hence the term vitrification. Vitrification can be applied to naked meristematic explants, such as shoot tips and embryogenic cells, or to propagules protected by encapsulation in alginate beads [200,201]. The most widely used vitrification solution is Plant Vitrification Solution 2 [202] and the method is droplet vitrification, where propagules are held on an aluminium strip covered in a droplet of vitrification solution and directly immersed in LN. However, the recently developed V cryo-plate technique, where droplet vitrification and encapsulation techniques are combined, has shown an improvement in recovery over droplet vitrification in some crops [203,204,205]. The two methods as applied to grapevine have been described by Bettoni et al. [206]. Another advantage of cryopreservation using vitrification is its ability to eliminate viruses, phytoplasma, and bacteria [169,207,208,209,210,211,212,213] for the delivery of high-health propagation material.

Through cryopreservation, viable explants can be brought to a state where cellular division and metabolic processes are minimized to the extent that they cease, preserving the structure and function of the biological system—virtually stopping the biological clock. There are no biochemical processes or gene expression; therefore, the genetic material is safe from any changes, and is hence ideal for conservation. Most of the protocols depend on a tissue culture phase (except the dormant bud technique used mainly for apple cryopreservation); thus, it is important to ensure the precautions mentioned in the previous section are taken to avoid somaclonal variation. It is also cost-effective to maintain collections for extended periods of time in LN compared with field or in vitro collections [214], and the cost effectiveness increases as more accessions are added to the collection [215]. LN is used to freeze material, so the method is not dependent on an electric power supply; hence, it is an attractive method for countries with an insecure power supply, although it is dependent on a reliable LN supply.

The use of cryobanking in genebanks using vitrification methods has recently been summarised by Wang et al. [200], and for the USDA, currently the world’s largest cryo collection, by Jenderek and Reed [216]. Panis [217] recently summarised the major cryopreserved collections worldwide. It is estimated that about 10,000 accessions are in long-term cryostorage using explants from in vitro culture material (mainly vitrification methods). Of these, over 80% belong to five species: potato (38%), cassava (22%), bananas and plantains (11%), mulberry (12%,) and garlic (5%). Other large collections are in apple, using winter dormant buds. Cryopreservation is also used for intermediate and recalcitrant seed crop conservation, such as coconut [218] and coffee [215], as well as for pollen. Pollen is naturally dehydration-tolerant and can be used to conserve the nuclear genetic diversity of CWR, recalcitrant species, endangered and rare species, and fruit and ornamental crops. The cryopreservation of pollen allows access to pollen when needed by breeders, particularly useful when breeding lines have differing flowering times, the use of CWR in breeding programmes, and for hybrid seed production programmes. Cryopreserving the pollen of male sterile lines for use on female lines can save large amounts of land in hybrid seed production fields and labour to collect pollen during busy periods of hybridisation. Pollen of many tropical plant species has successfully been cryopreserved: for example, the Indian Institute of Horticultural Research in Bangalore holds 650 samples of pollen from 45 species belonging to 15 plant families [219].

Synthetic seeds can be stored for long periods using vitrification-based cryopreservation [220,221]; both somatic embryos and other somatic tissues with meristematic regions have been used in cryopreservation and methods have been optimized to achieve post-thaw regeneration rates that meet genebank standards for the implementation of cryopreservation [201,206,222,223]. However, high genetic variability in the somatic embryogenesis response [224,225] is a barrier to use this propagule in cryopreservation.

Cryopreservation is only applied to a limited number of crops in some tropical genebanks outside the CGIAR system. Interestingly, the banana genebank, including the cryo-collection, is in Europe [226]. Since the introduction of vitrification methods, cryopreservation has become operationally simple and easily adaptable to any laboratory with tissue culture facilities. Collections of many tropical species, such as cassava, banana, sweet potato, and taro, are already being cryopreserved for their long-term security. The challenge is with CWR plus rare and endangered species of the tropics because their seeds are often recalcitrant and therefore cannot be preserved in traditional seed banks. Botanic gardens may only have one or a few specimens of each species in their field genebanks; however, the cryopreservation of seeds/embryonic axes and pollen enables the conservation of a much broader genetic diversity of CWRs or endangered species. Genebanks therefore need to invest in infrastructure and human resources for cryo-conservation. The integrated conservation strategies described above were recently used to save iconic New Zealand Myrtaceae species after the incursion of myrtle rust (*Austropuccinia psidii*) into New Zealand in 2017 [227]. The techniques used included the cryopreservation of pollen of *Metrosideros excelsa* and zygotic embryos of recalcitrant *Syzygium maire*, along with the successful hand pollination of flowers of *Metrosideros bartlettii*, a critically endangered species with only a few plants in the wild. Ex situ conservation strategies in field genebanks, in vitro slow growth, and the advantages and disadvantages of cryopreservation have recently been reviewed by Panis et al. [226].

## 7. Challenges to Plant Breeding in Search of Right Germplasm

Germplasm resources including CWRs carry alleles that are integral to creating new crop cultivars that can meet increasing consumer and environment demands. As changes in environmental conditions accelerate, so does the need for germplasm resources to breed more environmentally resilient crops. Properly characterised germplasm with information available to the plant breeding community will enable the deployment of improved cultivars on an ongoing basis. New cultivars with better water use efficiency, tolerance to soil toxicities, fertiliser response, and pest and disease tolerance will enable the replacement of at least some part of the costs for fertiliser, irrigation, and pesticides. Selection of the right genotype is the most effective means to achieve yield and quality improvements in crops, keeping other inputs to a minimum. Although plant breeders are often well aware of the necessity of maintaining genetic diversity in their breeding populations, they may lack the information to determine which of the thousands of accessions of a given crop would prove most beneficial for their breeding objectives. They may also be reluctant to introduce unadapted germplasm, with potentially negative impacts on quality, into their elite breeding materials. In some cases, they may lack the technical expertise or facilities to make interspecific crosses, for example, between CWR and cultivars of different ploidy levels that often require embryo rescue, in vitro pollination, and other interventions. Therefore, plant breeders should have access to facilities and expertise in plant biotechnology for the better utilisation of PGR.

Many international genebanks have a mandate to supply germplasm if the requests are justified. With more and more accessions being characterised and accessible databases made available, plant breeders should be trained in accessing the germplasm that matches their needs. The USDA-ARS holds over 576,000 accessions from 15,116 species and in 2015 alone, distributed over 239,000 accessions on request to national and international researchers [228]. The accessions available for supply can be searched in the Germplasm Resources Information Network-Global (GRIN-Global) database and ordered [229]. The CGIAR system of 11 genebanks holds over 736,000 accessions, with wheat and rice accounting for >196,000 and >144,000, respectively. Other major collections are in sorghum (>39,200), beans (~38,000), barley (>31,500), maize (>29,700), pearl millet (>23,000), and forages and fodder (>44,000). The CGIAR system distributed over 3.9 million samples over a 10-year period from 2007 to 2016 [143]. The CGIAR system of networks has developed a Global Information System (GLIS), with an emphasis of assigning a unique digital object identifier (DOI) to each accession; these are linked to the GRIN-Global searchable database and to Genesys—an online database for global plant collections managed by the Crop Trust [230]. The VIR collection holds 346,666 accessions of PGR and CWRs, representing 64 botanical families, 376 genera, and 2169 species. Annually, 12,000–14,000 accessions are distributed by the genebank, of which 2000–3000 accessions are supplied outside Russia [25].

## 8. Conclusions

Significant improvements have been made for the better management of PGRs within international and national genebanks in the last few decades. More attention is being paid to managing ‘difficult’ species with recalcitrant seeds, clonal crops, and CWRs. Effective research and development mechanisms and policies have been established for the protection and conservation of biodiversity worldwide. Many countries have signed international treaties on biodiversity, as well as PGR conservation and exchange. Coordination among genebank curators, breeders, farming communities, government organisations, and different stakeholders should be strengthened to meet global obligations for sustainable management and the use of PGR for food and nutritional security. Genebank accessions are of no use unless they are accessible online or can be observed in the field. Therefore, conserved PGRs need to be characterised using advanced technologies and results with passport data made available for online access by users. The emphasis should be on duplicating collections for better security, creating and securing core collections, identifying gaps in collections and remediating them, and using advanced and integrated strategies for the conservation and dissemination of information. CWR and landrace varieties need to be secured for the future, using multiple strategies including in situ, on-farm, and ex situ conservation. Further exploration and collection of PGR from diversity-rich centres should also assume priority, recognising the high rate of loss of genetic diversity.

## Figures and Tables

**Figure 1 plants-11-02038-f001:**
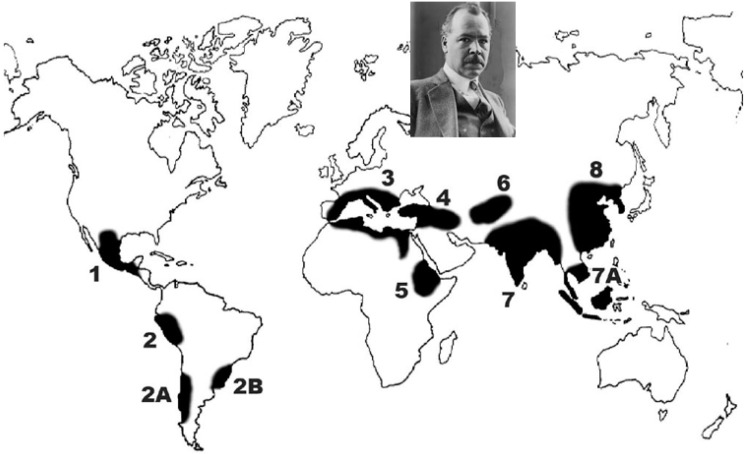
Eight main centres of the origin of cultivated plants according to Nikolai Vavilov (inset). 1. Mexico–Guatemala, 2. Peru–Ecuador–Bolivia, 2A. Southern Chile, 2B. Southern Brazil, 3. Mediterranean, 4. Middle East, 5. Ethiopia, 6. Central Asia, 7. Indo-Burma, 7A. Siam–Malaya–Java, and 8. China and Korea.

**Figure 2 plants-11-02038-f002:**
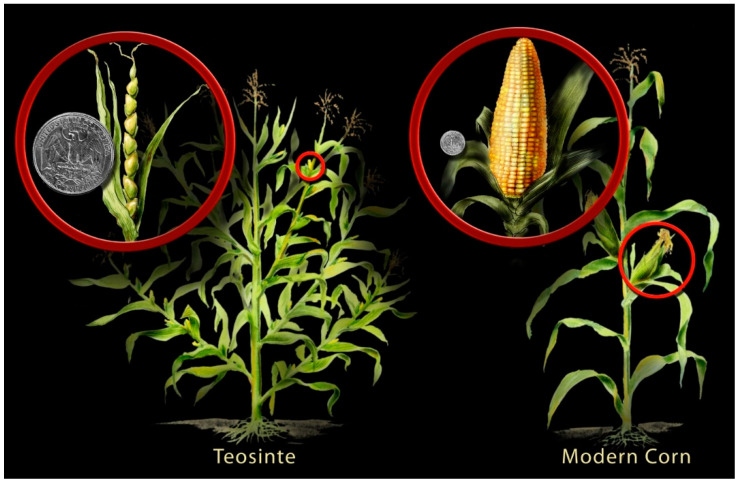
Domestication promotes rapid phenotypic evolution through artificial selection. Pictured here is wild grass teosinte (*Zea mays* ssp. *parviglumis*) that was domesticated into modern maize (*Z. mays* ssp. *mays*). The main traits selected during domestication included the ear and seed size (compared in relation to a USD coin in the inset) and the suppression of axillary branching. Figure courtesy National Science Foundation, USA.

**Figure 3 plants-11-02038-f003:**
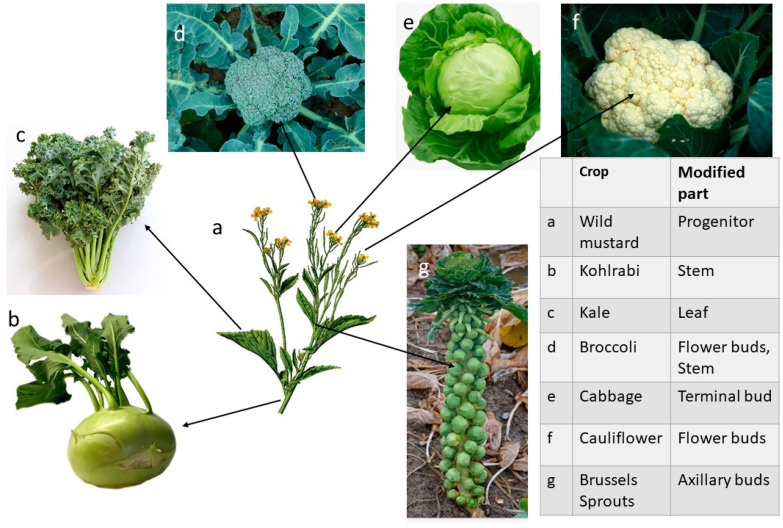
A whole range of Brassica vegetables have been selected during the diversification of *Brassica oleracea* (mustard), first domesticated as an oil-yielding crop in the Kurdistan/Mesopotamia area. Brussels sprouts are the youngest in the family of these vegetables, selected in Belgium in the mid-18th century.

**Figure 4 plants-11-02038-f004:**
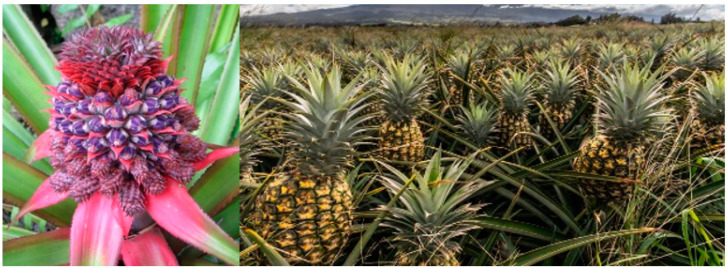
The difference in wild *Ananas bracteatus* (**left**) and domesticated pineapples *A. comosus* (**right**).

**Figure 5 plants-11-02038-f005:**
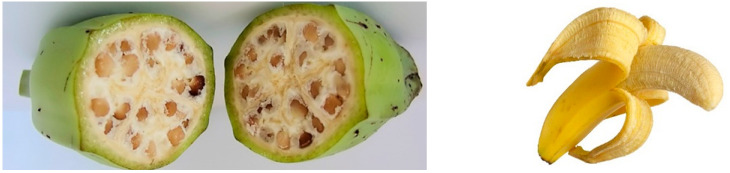
Wild banana is diploid and produces seeds (**left**), whereas cultivated banana is sterile and seedless because it is a triploid (**right**).

**Figure 6 plants-11-02038-f006:**
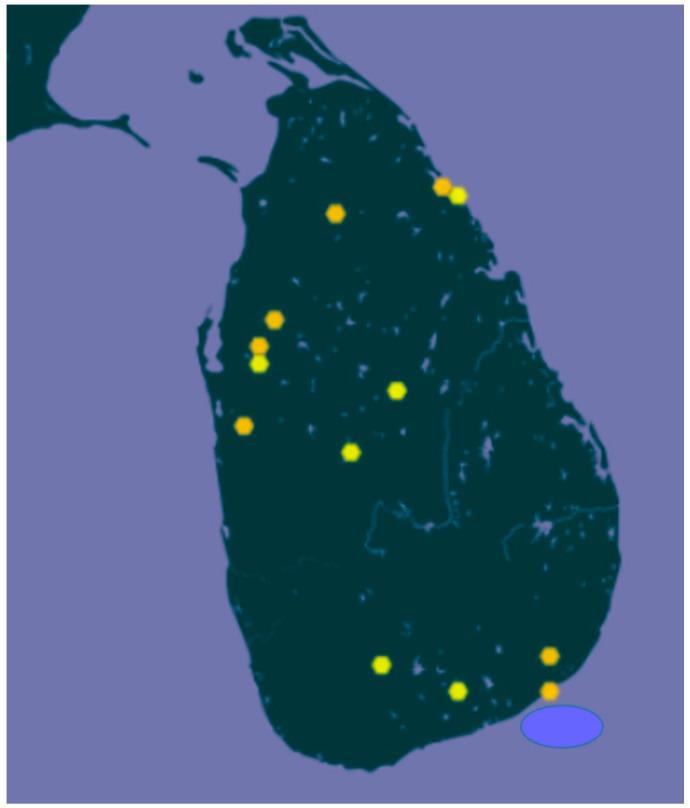
Distribution of *Oryza rhizomatis* discovered in the late 1980s in the periphery of Yala and Wilpattu National Parks in the driest areas of Sri Lanka [113]. Reproduced with permission from the Food and Agriculture Organization of the United Nations.

**Figure 7 plants-11-02038-f007:**
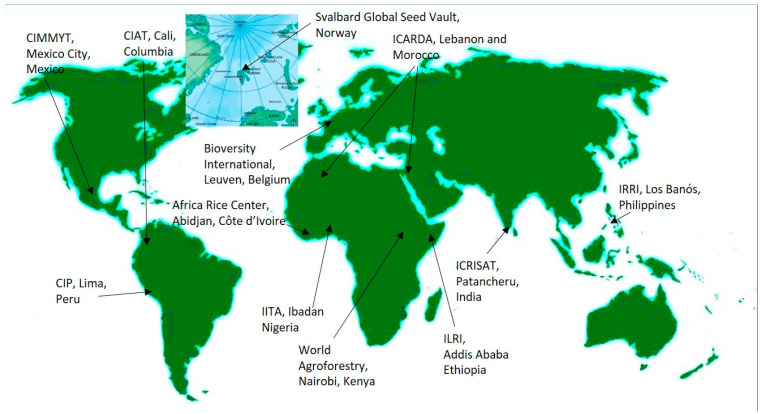
The location of the major international crop genebanks under the Consortium of International Agricultural Research Centres (CGIAR) network. CIMMYT—International Wheat and Maize Improvement Centre, CIAT—International Centre for Tropical Agriculture, ICARDA—International Centre for Agricultural Research, IRRI—International Rice Research Institute, ICRISAT—International Crops Research Institute for the Semi-Arid Tropics, ILRI—International Livestock Research Institute, IITA—International Institute for Tropical Agriculture, CIP—International Potato Centre. Inset—location of the Svalbard Global Seed Vault in the Arctic Circle.

**Figure 8 plants-11-02038-f008:**
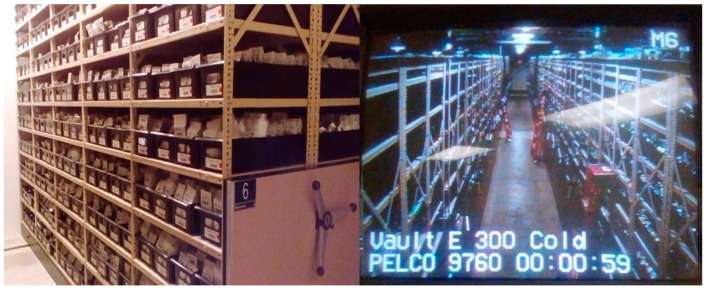
Seed vaults for long-term storage at the United States Department of Agriculture—Agricultural Research Service genebank in Fort Collins, Colorado.

**Table 1 plants-11-02038-t001:** Evolutionary timescale of life on land illustrating that crop domestication is a very recent event compared with evolution of land plants. mya—million years ago, ya—years ago. Data from multiple sources referred in text.

Time in History	Event
515–470 mya	First land plants
350 mya	Emergence of angiosperms
160 mya	Monocots separated from dicots
6.5 mya	Hominids appear
2 mya	*Homo habilis*
1.75 mya	*Homo erectus*
195,000–160,000 ya	*Homo sapiens*
130,000–120,000 ya	Human migration out of Africa
13,000 ya	Settled agriculture and beginning of crop domestication

**Table 2 plants-11-02038-t002:** Some of the crop species domesticated in different centres of diversity according to Zhukovsky [35].

Region	Crop
South Mexico–Central America	Avocado, Maize, Sweet Potato, Tomato, *Capsicum* spp., Tobacco, *Cucurbita pepo*, *C. moschata*, *Phaseolus* spp., *Amarnthus* *cruentus*, *A.* *hypochondriacus*, and *Gossypium hirsutum*
South American Andes (Peru, Bolivia, and Ecuador)	Potato, Quinoa, Lima Bean, Common Bean, Tomato, *Capsicum* spp., *Cucurbita maxima, C. moschata,* Grain amaranth (*Amaranthus caudtus*), Oca (*Oxalis tuberosa*), Ulluco (*Ullucus tuberosus*), Añu (*Tropaeolum tuberosum*), Achira (*Canna edulis*), Coca, *Gossypium barbadense*
Tropical lowland South America (Chile, Paraguay, and Southern Brazil)	Cassava, Arrowroot, Cocoyam, Peanut, Pineapple, and *Capsicum chinense*
Mediterranean	Grapevine, Carrot, Cabbage, Olive, Sugar Beet, European Pear, *Vicia faba*, *V. sativa*, *Lathyrus ochrus*, *Cicer arietinum*, and Almond
Asia Minor (Middle East)	*Cicer arietinum* (secondary centre), *Lens culinaris, L. orientalis*, *Vicia ervilia, Pisum sativum, Medicago sativa*, *Trifolium resupinatum*, *Trigonella foenum-graecum*, *Onobrychis* spp., *Lathyrus cicera*, *Vicia* spp., Date Palm, and Lettuce
Abyssinia (Ethiopian Centre)	Millets, Sorghum, Castor, Coffee (*Coffea arabica* L.), Peanut, Teff (*Eragrostis abyssiniaca* Link.), Finger Millet, Sesame, and Niger (*Guizotia abyssiniaca* Cass.)
Inner Asia	Wheat, Barley, Apple, and Onion
India	Mung Bean, Rice, Black Gram, Pigeon Pea, Horsegram, Mango, Little Millet (*Panicum sumatrense*), and Flax
Indo-Malaya	Rambuttan, Banana, Sugarcane, and Yam
China	Rice, Soybean, Peach, Foxtail Millet (*Setaria italica*), Proso Millet (*Panicum miliaceum*), Hemp, Tea, Chinese Cabbage, Mulberry, and *Citrus* spp.

**Table 3 plants-11-02038-t003:** Some examples of traits of wild rice used in improving *Oryza sativa*—the cultivated species of rice.

*Oryza* Species	Genome	Trait of Interest	Line Number	Reference
*O. glaberrima*	AA	Brown planthopper (*Nilaparvata lugens* Stål.) resistance	IR 75870-5-8-5-B-1-B), IR 75870-5-8-5-B-2-B)	[117]
*O. nivara*	AA	Brown planthopper tolerance	IR28, IR29, IR30, IR34, IR36, IR38, IR40, IR48, IR50, IR56, IR58,	[118]
*O. minuta*	BBCC	Brown plant hopper resistance	IR 71033-62- 15, IR 71033-121-15	[117]
*O. nivara*	AA	Sheath blight (*Rhizoctonia solani*) tolerance	RPBio4918-10-3	[119]
*O. nivara*	AA	Salinity tolerance	14S, 75S, 166S, IL 3-1K	[120]
*O. rufipogon*	AA	Salinity tolerance	Chinsurah Nona 2	[118]
*O. nivara*	AA	Heat tolerance	166-2, 175-2, 3-1K	[121]
*O. rufipogon*	AA	Heat tolerance	377-13, 50	[121]
*O. nivara*	AA	Heat tolerance	24S, 70S, 14-3S	[122]
*O. nivara*	AA	High yield	220S, 10-2S	[123]
*O. nivara*	AA	100 grain weight, early flowering	NSL-15, NSL-22	[124]
*O. sativa* f. *spontanea*	AA	Cytoplasmic male sterility		Mondal and Henry [118]
*O. rufipogon*	AA	Rice tungro bacilliform virus resistance	Matatag 9	[118]
*O. longistaminata*	AA	Bacterial blight (*Xanthomonas oryzae* pv. *oryzae*)	Shanyou63-*Xa21*	[125]
*O. rufipogon*	AA	Acid sulphate tolerance	AS 996	[118]
*O. minuta*	BBCC	Bacterial blight	41 Lines	[126]
*O. minuta*	BBCC	Brown planthopper	11 Lines	[126]
*O. minuta*	BBCC	Whitebacked planthopper (*Sogatella furcifera*)	7 Lines	[126]
*O.* *grandiglumis*	CCDD	Grain weight and other yield traits	HG 101	[127]
*O. meridionalis*	AA	Iron tolerance	CM 23, CM 24	[128]
*Oryza rufipogon* ‘DXWR’	AA	Drought tolerance	Restorer line BIL627	[129]

**Table 4 plants-11-02038-t004:** Genebanks with large in vitro collections.

Genebank	Country	Crop	Number of Accessions	Reference
International Potato Centre	Peru	Potato, Andean Root and Tubers	>11,000	CIP-Genebank [190]
International Institute for Tropical Agriculture	Nigeria	Cassava	>2500	IITA-GRC [191]
		Yam	>2500	
		Banana	>500	
EMBRAPA Genebank	Brazil	24 genera, 63 species	1250	Cunha Alves, et al. [192]
Agricultural Research Council	South Africa	Potato	1100	Myeza and Visser [193]
National Bureau of Plant Genetic Resources	India	Fruit crops	743	Tyagi and Agrawal [194]
		Tuber crops	611	
		Spices	380	
		Bulbous crops	171	
		Medicinal and Aromatic	170	
		Total 24 Genera, 63 spp.	1250	
Bioversity International Transit Centre	Belgium	*Musa* spp.	>1500	ITC [195]
International Centre for Tropical Agriculture (CIAT)	Columbia	Cassava	6632	Rondon [186]
The New Zealand Institute for Plant and Food Research Limited	New Zealand	Kiwifruit (*Actinidia* spp.)	1012	Debenham and Pathirana [62]
United States Department of Agriculture Agricultural Research Service	USA	Potato	~1000	Bamberg et al. [196]

## Data Availability

Not applicable.

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
