# Peer review of "Management and Utilization of Plant Genetic Resources for a Sustainable Agriculture"

_plants, 2022, doi:10.3390/plants11152038_

Round 1

Reviewer 1 Report

The paper is a comprehensive study on the state of global PGR and the authors cited relevant literature. I only have 2 suggestions for improvement.

1. The last paragraph (lines 1157-1169) can be revised or even removed. I enjoyed reading the whole paper, until I reached this point. It sounded more depressing than necessary. While industrial agriculture has contributed to SDG2, we have already gone a long way from this narrative. The authors could have ended it at line 1156.

2. On the discussion of ex situ conservation and management, you can mention work on safety duplication at Svalbard Global Seed Vault to update the landscape.

Minor edits:

line 772 plant treaty > Plant Treaty

Fig 7. The genebank of Bioversity is located in Leuven, Belgium

Author Response

File attached

Reviewer 2 Report

This manuscript is an ambitious attempt to synthesize the origin, conservation and use of plant genetic resources. This is a very broad topic, making it a difficult challenge.

The efforts of the authors should be acknowledged, but I consider that the objective is not reached. The manuscript is composed of different parts, of unbalanced size and interest, without much connection between them. For example, the part on domestication is interesting in itself, but how does it serve the general purpose? Why so much importance given to core collections or cryopreservation ?

There are also some problems with the structure of the text. For example, in section 4.2 "Approaches to germplasm conservation", there are sub-sections on in situ and ex situ conservation, as expected, but also on "perennials crops", without it being clear why the conservation of these perennials crops do not fall under in or ex situ conservation.

I suggest that the purpose of this synthesis be better defined by emphasizing its most original character : "How should the conservation and use of plant genetic resources advance to better serve agricultural sustainability?" It seems to me that this question is close to the one the authors intended to deal with (Lines 113-115), but that it was unfortunately lost during the writing of the manuscript. For example, the part on ex situ conservation does not suggest much changes in the ways to build, manage and evaluate collections and make them more useful to address agroecological challenges. I don't doubt that germplasm conservation can contribute to sustainable agriculture, but the article doesn't provide many new ideas or identify new avenues to stimulate this contribution. I was expecting more discussions like the one on new perennial crops, which is interesting.

It is unfortunate that the manuscript does little to address the social and political issues associated with the conservation and use of genetic resources. For example, suggesting the use of genome editing technologies (L709) is relevant from a purely biological point of view, but the social acceptability of these technologies is far from certain.

Here are more specific remarks :

L1. The title does not reflect properly the content of the paper.

L20. « agrobiodiversity » encompasses more than crop genetic resources.

L23-L25. Engaging the civil society in coillaboration with genebanks is an interesting issue, but it is poorly adressed in the paper.

L68. Add a reference on the commitments were done in 2010 (Aïchi targets ?)

L71. « co-coordinated » : by whom ?

L73-74. There is no convention on Climate change and Biodiversity. There is one on climate change and another one on biological diversity.

L126. This part is much too long. Does the theory of homologous series of variation deserve so much details ?

L128. An update on the knowledge on domestication centers (e.g. SCARCELLI, Nora, CUBRY, Philippe, AKAKPO, Roland, et al. Yam genomics supports West Africa as a major cradle of crop domestication. Science advances, 2019, vol. 5, no 5, p. eaaw1947) would be more useful that the usual reminder on Vavilov’s theories.

L393. This part (4.1) lacks references to support the overall assumption of genetic erosion.

L514. This part (4.3.3) on the implementation of in situ conservation is rather poor. Much research has been done on the subject. For on-farm conservation, more ideas in : JARVIS, Devra I., HODGKIN, Toby, STHAPIT, Bhuwon R., et al. An heuristic framework for identifying multiple ways of supporting the conservation and use of traditional crop varieties within the agricultural production system. Critical Reviews in Plant Sciences, 2011, vol. 30, no 1-2, p. 125-176.

L703-704. Beyond the issue of new perennial crops, the topic of neo-domestication should be addressed in this paper (see for example TOMOOKA, Norihiko, NAITO, Ken, KAGA, Akito, et al. Evolution, domestication and neo-domestication of the genus Vigna. Plant Genetic Resources, 2014, vol. 12, no S1, p. S168-S171. ; ZHAO, Yao, ZHONG, Lan, ZHOU, Kai, et al. Seed characteristic variations and genetic structure of wild Zizania latifolia along a latitudinal gradient in China: implications for neo-domestication as a grain crop. AoB Plants, 2018, vol. 10, no 6, p. ply072. ; …)

L783. Figure 7 is of low quality. What it its source ? Does ILRI maintain crop germplasm collections ?

L785-786. « CGIAR » is no longer an acronym

L866-869. This level of detail is not useful in a review paper. This is just one example among many in the manuscript.

Author Response

File attached

Reviewer 3 Report

The authors of the manuscript have prepared a broad and detailed review of the conservation of genetic resources. It contains information about the activities of genetic resources taking place in the world, their types, systems, and preserved accessions. Therefore, the manuscript can be considered a good review of the current situation of genetic resources.

However, it should be noted that according to the title of the manuscript, a little different information was expected - not a review of the existing types of genetic resources, systems, etc., but more analysis of that genetic diversity (research done on this diversity) is found and available in germplasm collections. Therefore, I would advise the authors to rethink the title of the manuscript. The same applies to the Abstract, after reading it, a bit different information was expected. The manuscript includes quite a lot of general information that is socially relevant but not necessarily related to crop genetic diversity. In my opinion, removing such information would only improve the scientific quality of the manuscript.

The same can be said about the generally known scientific information provided, which does not need to be described in detail, references are enough (for example, a long description of Vavilov's works, and centres of cultivated plant origin). This is already textbook information that does not need to be repeated in a scientific publication.

In order to improve the existing manuscript, the authors should choose which topic they want to develop: 1) review of the current situation in the conservation of genetic resources - types, collections, networks, and methods (as evidenced by the main text of the manuscript, which is very well described and provides good information ); 2) review of the current genetic diversity of germplasm in the collections (as evidenced by the title and abstracts, but minimally mentioned in the text - only in the introductory part in connection with the history of domestication). By making these changes, the manuscript would be relevant and contribute to scientific knowledge.

Author Response

File attached

Round 2

Reviewer 2 Report

I thank the authors for having taken many of my comments into account. The revised manuscript has been significantly improved.

I made a very few comments in the attached file, suggesting minor changes.

Author Response

Dear Reviewer,

Thank you for the critical reading of the revised manuscript and suggesting minor improvements. In line 78 (PDF) we changed the word convention to conventions, in line 79, we left the sentence after saying that it is a draft and it now reads "This new draft for post-2020 Global Biodiversity Framework comprises 21 targets and 10 milestones [8]." In line 84, we have given a reference from PNAS for the statement that the Green Revolution resulted in increased micronutrient deficiencies. To provide the positive side of the Green Revolution, we changed the sentence (using another reference from PNAS) to: "Although the increased yield helped to save the cultivation of an estimated 17.9 – 26.7 million hectares of new land under crops [9], the resulting increase in food production during the past five decades was accompanied by environmental degradation and deficiency in micronutrients in populations [10]." Sentence in line 676 has been modified to: "In general, implementing in situ conservation has been more challenging than ex situ conservation for several reasons".

Thank you again 

Reviewer 3 Report

Thanks to the authors for the corrections! It is understandable that the topic is so broad that it is not possible to include all possible aspects of PGR in one review. But in the current version of the manuscript, it is well done and the information is publishable.

Author Response

Dear Reviewer,

We appreciate your time to check the first revision and accepting the manuscript in the present form.